# Turbulent superstructures in Rayleigh-Bénard convection

Ambrish Pandey [1], Janet D. Scheel [2] & Jörg Schumacher [1]

Turbulent Rayleigh-Bénard convection displays a large-scale order in the form of rolls and cells on lengths larger than the layer height once the fluctuations of temperature and velocity are removed. These turbulent superstructures are reminiscent of the patterns close to the onset of convection. Here we report numerical simulations of turbulent convection in fluids at different Prandtl number ranging from 0.005 to 70 and for Rayleigh numbers up to $10^7$. We identify characteristic scales and times that separate the fast, small-scale turbulent fluctuations from the gradually changing large-scale superstructures. The characteristic scales of the large-scale patterns, which change with Prandtl and Rayleigh number, are also correlated with the boundary layer dynamics, and in particular the clustering of thermal plumes at the top and bottom plates. Our analysis suggests a scale separation and thus the existence of a simplified description of the turbulent superstructures in geo- and astrophysical settings.

[1] Institut für Thermo- und Fluiddynamik, Technische Universität Ilmenau, Postfach 100565, D-98684 Ilmenau, Germany. [2] Department of Physics, Occidental College, 1600 Campus Road, M21, Los Angeles, CA 90041, USA. Correspondence and requests for materials should be addressed to J.S. (email: joerg.schumacher@tu-ilmenau.de)

Large temperature differences across a horizontally extended fluid layer induce a turbulent convective fluid motion which is relevant in numerous geo- and astrophysical systems[1]. These flows are typically highly turbulent with very large Rayleigh numbers Ra, the parameter that quantifies the intensity of the thermal driving in convection. From the classical perspective of turbulence one would expect a chaotic, irregular motion of differently sized vortices and thermal plumes. Rather than such a featureless stochastic fluid motion, some turbulent flows in nature display an organization into prominent and regular flow patterns that persist for times longer compared to an eddy turnover time and extend over lengths which are more than the characteristic height scale. Examples are cloud streets in the atmosphere[2] or granulation networks at the solar surface[3] and other stars[4]. Large-scale order that is greater than the height of the convection layer will be termed a turbulent superstructure for Rayleigh-Bénard convection. It thus extends a notion that originally stands for fluctuation patterns in wall-bounded shear flows[5]. Super-structures are observed in turbulent convection flows with very different molecular dissipation properties. The Prandtl number $Pr = \nu/\kappa$, another dimensionless parameter which relates kinematic viscosity $\nu$ to temperature diffusivity $\kappa$, is for example very small for stellar convection, $Pr \lesssim 0.001$[6–8]. It is 0.7 for atmospheric flows and 7.0 for heat transport in the oceans. Rayleigh-Bénard convection (RBC) is the simplest turbulent convection flow evolving in a planar fluid layer of height $H$ that is uniformly heated with a temperature $T = T_b$ from below and cooled from above with $T = T_t$ such that $T_b - T_t = \Delta T > 0$. The Rayleigh number is given by $Ra = g\alpha\Delta T H^3/(\nu\kappa)$ with $g$ being the acceleration due to gravity and $\alpha$ the thermal expansion coefficient. RBC can be considered as a paradigm for many applications[9, 10] that usually contain further physical processes, such as radiation[11] and phase changes[12, 13], and additional fields such as magnetic fields[14]. Numerical simulations of convection[15–21] have enabled researchers to access the large-scale structure formation in turbulent convection flows. Long-term numerical investigations at very small Prandtl numbers $Pr \ll 0.1$ for time spans of the order of 10 or more turnover times require simulations on massively parallel supercomputers in order to resolve the highly inertial turbulence properly (a turnover time is defined to be $\tau/3$ where $\tau$ is defined later in the text.) Such simulations have not been done before and this is a central motivation for the present study.

At the onset of convection, $Ra_c = 1708$, straight convection rolls have a unique and Prandtl-number-independent wavelength, $\lambda_c \approx 2H$[22, 23]. For $Ra \gtrsim Ra_c$, these rolls become susceptible to secondary linear instabilities causing modulations, such as Eckhaus, zig-zag or oscillatory patterns[24–26]. These secondary instabilities depend strongly on the Prandtl number of the working fluid and the wavenumber range of the plane-wave perturbation to the straight convection rolls in the layer[24]. Dependencies on Rayleigh and Prandtl numbers of the pattern wavelength for $Ra > Ra_c$ have been studied systematically in RBC experiments in air, water and silicone oil by Willis et al[27]. Average roll widths tend to increase with Ra, which the authors attributed to increasingly unsteady three-dimensional motions. The trend with growing Pr is less systematic[15] and accompanied by hystereses at $Pr \gg 1$[27].

Roll and cell patterns of the velocity field in a turbulent RBC for $Ra \gtrsim 10^5$ that are reminiscent of the flow structures in the weakly nonlinear regime at $Ra \lesssim 5 \times 10^3$ have been observed in recent DNS at $Pr \gtrsim 1$[19, 20]. Their detection requires an averaging over a time interval that should be long enough to remove the turbulent fluctuations in the fields effectively and yet short enough to not wash away the large-scale structures[20]. A sliding time average with an appropriate time window width should thus be able to separate the fast, small-scale turbulent fluctuations of velocity and temperature from the gradual variation of the large-scale superstructure patterns. The determination of this averaging time scale as a function of Ra and Pr is a second motivation for the present study.

In the present work, we report an analysis of the characteristic spatial and temporal scales of turbulent superstructures in RBC by means of three-dimensional direct numerical simulations (DNS) spanning more than four orders of magnitude in Pr and more than three orders in Ra. We identify the characteristic averaging time scales, $\tau(Ra, Pr)$, which will be connected with a characteristic spatial scale (or wavelength) that can be determined by a spectral analysis of the turbulent superstructures. Our study of large-aspect-ratio turbulent RBC extends to very small Prandtl numbers with values significantly below 0.1, which have not been obtained before. We also clearly demonstrate the gradual evolution of the turbulent superstructures at all Prandtl numbers. It is done by radially averaged, azimuthal power spectra that reveal a gradual switching of the orientation of the superstructures which is reminiscent of cross-roll or skewed varicose instabilities that are well-known from the weakly nonlinear regime of RBC. Furthermore, we compare the characteristic pattern scale in the bulk of the RBC flow to the scales of plumes and plume clusters that are present in the boundary layers in the vicinity of the top and bottom walls. The temperature patterns in the bulk are found to be correlated with the most prominent ridges of the derivative of the temperature field with respect to the vertical coordinate (at the bottom and top plates) which in turn are correlated with the wall stresses of the advecting velocity. Our analysis provides characteristic separation time and length scales for turbulent convection flows in extended domains and thus opens the possibility to describe the superstructure patterns in turbulent convection by effective and reduced models that separate the fast, small scales from the slow, large scales. These reduced models can advance our understanding of a variety of turbulent systems that exhibit large-scale pattern formation, including mesoscale convection and solar granulation.

## Results

**Superstructure patterns.** All simulations reported here are of the Boussinesq equations of motion and performed in an extended closed square cell of aspect ratio of 25:25:1. Further details are found in Table 1 and the methods section. Figure 1 shows the velocity field lines (top row) and the corresponding temperature contours in the midplane (bottom row) for a simulation at one of our lowest Prandtl numbers. While the instantaneous pictures display the expected irregularity of a turbulent flow as visible for example by the streamline tangle in Fig. 1a, the averaged data reveal a much more ordered pattern. We also see that the superstructure patterns are more easily discerned in temperature field snapshots than in those of the velocity field. Figure 2 confirms this observation. Here, we plot the root mean square (rms) values of the vertical velocity component $u_z$ and the temperature $T$. In agreement with Fig. 1, we split both fields into contributions coming from the time average over the time interval $\tau$ and the fluctuations,

$$u_z(\boldsymbol{x}, t) = U(\boldsymbol{x}) + u_z'(\boldsymbol{x}, t), \tag{1}$$

$$T(\boldsymbol{x}, t) = \Theta(\boldsymbol{x}) + T'(\boldsymbol{x}, t). \tag{2}$$

The averaging volume $\tilde{V}$ is a slab around the midplane. See Eqs. (4) and (5) later in the text for definitions of $U$ and $\Theta$. It can be seen that the rms values of the total and time averaged temperature are always close together when Prandtl and Rayleigh

**Table 1 Parameters of the different spectral element simulations**

|    | Pr    | Ra            | $N_e$     | N  | Nu          | Re         | $u_{rms}$        | $N_s$ | $\tau_{total}$ | $\tau$ | $t_\kappa$ | $t_\nu$ |
|----|-------|---------------|-----------|----|-------------|------------|------------------|-------|--------|------|-----|------|
| 1  | 0.005 | $10^5$        | 2,367,488 | 11 | 1.9 ± 0.01  | 2491 ± 20  | 0.56 ± 0.004     | 63    | 60     | 21   | 22  | 4472 |
| 2  | 0.021 | $10^5$        | 2,367,488 | 7  | 2.6 ± 0.01  | 1120 ± 8   | 0.51 ± 0.004     | 69    | 156    | 27   | 46  | 2182 |
| 3  | 0.7   | $10^5$        | 2,367,488 | 3  | 4.2 ± 0.02  | 92 ± 0.4   | 0.24 ± 0.001     | 117   | 1159   | 57   | 265 | 378  |
| 3a | 0.7   | $10^5$        | 1,352,000 | 5  | 4.3 ± 0.02  | 92 ± 0.4   | 0.24 ± 0.001     | 278   | 1108   | 57   | 265 | 378  |
| 4  | 7     | $10^5$        | 2,367,488 | 3  | 4.1 ± 0.01  | 11 ± 0.04  | 0.09 ± 0.001     | 150   | 2979   | 192  | 837 | 120  |
| 4a | 7     | $10^5$        | 1,352,000 | 5  | 4.1 ± 0.01  | 11 ± 0.03  | 0.09 ± 0.001     | 268   | 2670   | 207  | 837 | 120  |
| 5  | 35    | $10^5$        | 1,352,000 | 5  | 4.5 ± 0.01  | 2.3 ± 0.005| 0.04 ± 0.0001    | 343   | 3420   | 300  | 1871| 53   |
| 6  | 70    | $10^5$        | 1,352,000 | 5  | 4.6 ± 0.01  | 1.2 ± 0.002| 0.03 ± 0.0001    | 337   | 3360   | 375  | 2646| 38   |
| 7  | 0.7   | $2.3 \times 10^3$ | 2,367,488 | 3  | 1.3 ± 0.001 | 6.1 ± 0.04 | 0.11 ± 0.001     | 74    | 1460   | 93   | 40  | 57   |
| 8  | 0.7   | $5.0 \times 10^3$ | 2,367,488 | 3  | 1.8 ± 0.005 | 14 ± 0.2   | 0.17 ± 0.002     | 51    | 250    | 54   | 59  | 85   |
| 9  | 0.7   | $10^4$        | 1,352,000 | 5  | 2.2 ± 0.01  | 24 ± 0.2   | 0.20 ± 0.002     | 240   | 1195   | 72   | 84  | 120  |
| 10 | 0.7   | $10^6$        | 2,367,488 | 7  | 8.1 ± 0.03  | 290 ± 1    | 0.24 ± 0.001     | 392   | 1292   | 66   | 837 | 1195 |
| 11 | 0.7   | $10^7$        | 2,367,488 | 11 | 15.5 ± 0.06 | 864 ± 3    | 0.23 ± 0.001     | 127   | 248    | 72   | 2646| 3780 |

We display the Prandtl number Pr, the Rayleigh number Ra, the number of spectral elements $N_e$, the polynomial order of the expansion in each of the three space directions $N$ on each element, the Nusselt number Nu, the Reynolds number Re. Their definitions are given by Eqs. (13) and (14) in the Methods section. Furthermore, the root mean square velocity $u_{rms} = \langle u_i^2 \rangle_{V,t}^{1/2}$, the number of statistically independent snapshots $N_s$, and the total runtime $\tau_{total}$ in units of the free-fall time $T_f = H/U_f$ are shown. We also list the turnover time $\tau$, the vertical diffusion time $t_\kappa = H^2/\kappa = \sqrt{RaPr}T_f$, and the vertical viscous time $t_\nu = H^2/\nu = \sqrt{Ra/Pr}T_f$. The snapshot separation is found from $\tau_{total}/(N_s - 1)$ and $t_0$ is given in these units. Runs 3a and 4a are conducted at different spectral resolution and compared with runs 3 and 4, respectively

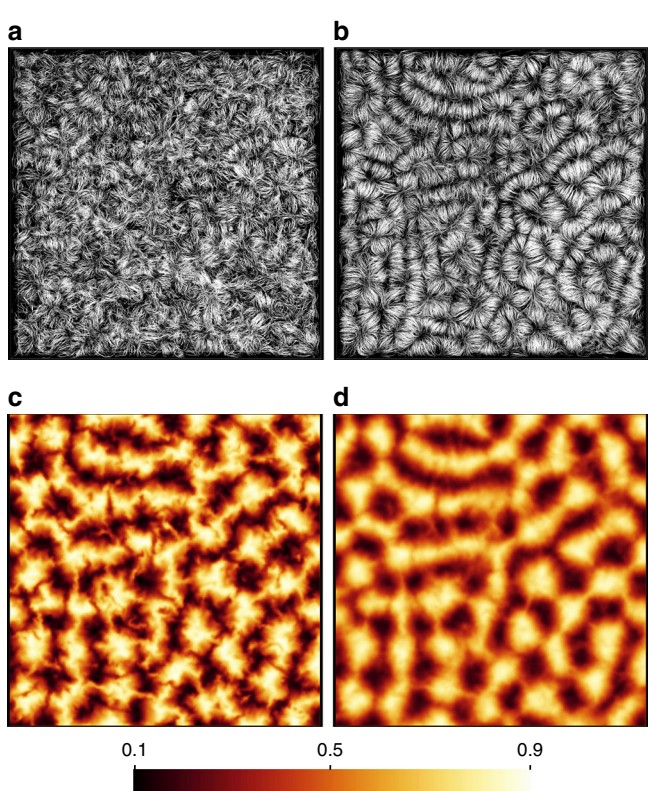

**Fig. 1** Instantaneous and time-averaged fields. Field line plots of the instantaneous (**a**) and time-averaged (**b**) velocity for Pr = 0.021 and Ra = $10^5$. View is from the bottom. Corresponding instantaneous (**c**) and time-averaged (**d**) temperature field in midplane. Averaging time is 27 free-fall times ($T_f$). The three-dimensional simulation domain is resolved by more than 1.2 billion mesh cells

numbers are varied. This is in contrast to $u_z$. Fluctuations dominate here when the Prandtl numbers are low and the Rayleigh numbers are sufficiently high. Time averaging is thus necessary to reveal the patterns for both turbulent fields.

Figure 3 displays velocity field lines and temperature contours of time-averaged turbulent RBC flows at Prandtl number ranging from Pr = 0.005 to 70 at Ra = $10^5$ and at Rayleigh number ranging from Ra = $5 \times 10^3$ to $10^7$ for convection in air at Pr = 0.7. All runs are chaotically time-dependent and show non-vanishing velocity fluctuations about the time-averaged fields. We also refer to Supplementary Fig. 1 and Supplementary Note 1 for vertical mean profiles of velocity fluctuations. In most cases, the magnitude of these velocity fluctuations exceeds the value of the mean flow (see Fig. 2) and the system is thus fully turbulent. In cases (e) at Ra = $10^5$, Pr = 70, (f) at Ra = 5000, Pr = 0.7, and (g) at Ra = $10^4$, Pr = 0.7 this is not the case and we denote these flows as weakly turbulent. For the weakly turbulent cases the time averaged data does not deviate significantly from the instantaneous snapshots. If we look at the trends for all runs, we see that the velocity field lines form curved rolls for the lower Pr and cell-like patterns for Pr ≥ 7. These structures fill the whole layer and are reminiscent of patterns at the onset of convection at much smaller Rayleigh numbers[26]. The corresponding temperature averages in the midplane show alternating ridges of cold downwelling and hot upwelling fluid which are coarser for the lowest Prandtl numbers and the highest Rayleigh numbers, respectively. For Pr = 0.005 and 0.021, this is due to the highly diffusive temperature field that is in conjunction with an inertia-dominated fluid turbulence[28–30]. In case of the highest Prandtl number, Pr = 70 at Ra = $10^5$, the amplitude of the turbulent velocity field fluctuations is significantly smaller and the temperature field displays much finer filaments. Coarser temperature patterns can also be observed for the highest Rayleigh number at Ra = $10^7$. While low-Prandtl-number convection transports momentum very efficiently, the heat transport becomes significantly larger at the higher Prandtl numbers as seen in Table 1. Figure 3 also demonstrates that the characteristic mean width of the rolls and spirals varies with Pr and Ra.

**Characteristic times and scales.** The free-fall time, $T_f = (H/g\alpha\Delta T)^{1/2}$ is a characteristic convective time unit that stands for the (relatively) fast dynamics of thermal plumes and vortices in a turbulent convection flow. It differs from the turnover time, i.e., the time it takes for a fluid parcel to circulate in a convection roll. A slower time unit in the turbulent flow is either a vertical viscous (Pr < 1) or a vertical diffusive (Pr > 1) time composing an effective dissipative time by $T_d = \max(t_\nu, t_\kappa)$ with $t_\nu = H^2/\nu$ and $t_\kappa = H^2/\kappa$.

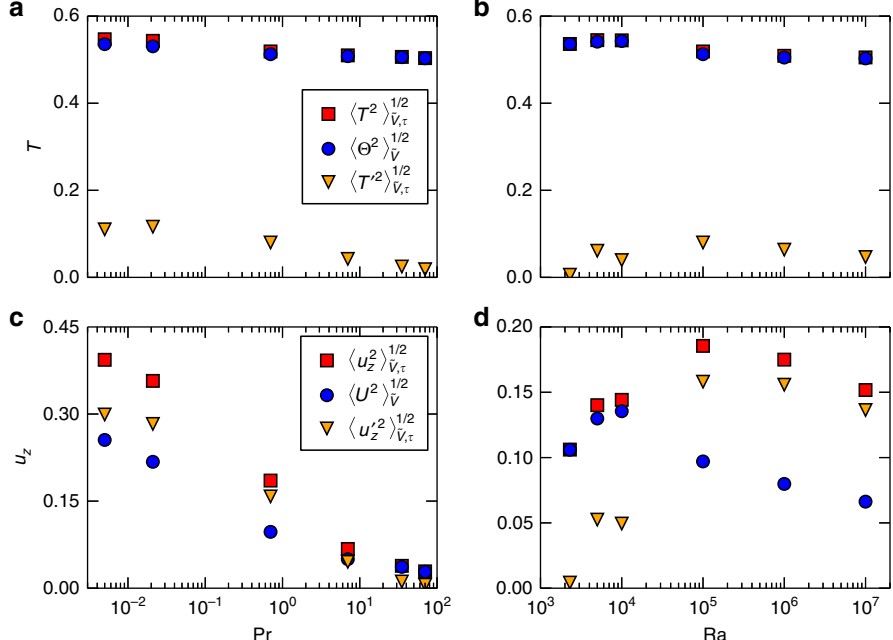

**Fig. 2** Magnitude of turbulent superstructures. The panels compare the root mean square (rms) values of the full field, $\langle T^2 \rangle^{1/2}_{\bar{V},\tau}$ and $\langle u_z^2 \rangle^{1/2}_{\bar{V},\tau}$, the temporal means $\langle \Theta^2 \rangle^{1/2}_{\bar{V}}$ and $\langle U^2 \rangle^{1/2}_{\bar{V}}$ as defined in Eqs. (4, 5), and the fluctuations about the temporal mean, $\langle T'^2 \rangle^{1/2}_{\bar{V},\tau'}$ and $\langle u_z'^2 \rangle^{1/2}_{\bar{V},\tau'}$, for the temperature $T$ in **a**, **b** and for the vertical velocity component $u_z$ in **c**, **d**. The dependence on Pr at Ra = $10^5$ is given in **a**, **c** and on Ra at Pr = 0.7 in **b**, **d**. The data for the temporal means are shown in Fig. 3

A complete removal of the large-scale patterns would require an averaging period of multiples of the horizontal dissipation time $\Gamma^2 T_d$ with $\Gamma$ being the aspect ratio of the domain[31, 32]. This time scale is the largest one that can be composed in the system by dimensional arguments. For the present cases at hand, this horizontal dissipation time is larger than $10^4 T_f$ and covers thus time spans which are not accessible in our massively parallel turbulence simulations.

Thus, the averaging time $\tau$ that separates small-scale turbulence and superstructures should be bounded by

$$T_f \ll \tau(\text{Ra}, \text{Pr}) \ll T_d. \qquad (3)$$

This time $\tau$ should be considered as a representative value of a finite range of times rather than an exact time and is expected to obey a dependence on our two system parameters, Ra and Pr. In Supplementary Figs. 2 and 3 and Supplementary Note 2 it is shown for two different Prandtl numbers how the patterns change when the averaging time is varied. On the one hand, $\tau$ should be long enough to remove all small-scale fluctuations and to reveal the superstructures, in particular of velocity. On the other hand, $\tau$ has to be short enough such that the large-scale patterns are not removed completely. Hence we define $\tau$ as the characteristic turnover time of fluid parcels in the circulation rolls or cells, the latter of which extend across the whole layer from bottom to top and are considered as the building blocks of the superstructure velocity patterns.

We decompose the RBC fields into a fast changing and gradually evolving contribution. This is inspired by asymptotic expansions that are developed for constrained turbulence[33–35]. Furthermore, we substitute the full temperature field, $T(\boldsymbol{x}, t)$, by its deviation from the linear diffusive equilibrium profile, $\theta(\boldsymbol{x}, t) = T(\boldsymbol{x}, t) - T_{\text{lin}}(z)$ for the following Fourier analysis. Our focus is on the horizontal patterns in the system. Therefore, the subsequent superstructure

analysis is focussed on the symmetry plane at $z = 1/2$ where the patterns are identified by upwelling hot and downwelling cold fluid (see Fig. 4a). The gradually varying fields are given by the following sliding time average with respect to $\tau$

$$U(x, y; \tau, t_0) = \frac{1}{\tau} \int_{t_0 - \tau/2}^{t_0 + \tau/2} u_z(x, y, z = 1/2, t')dt', \qquad (4)$$

$$\Theta(x, y; \tau, t_0) = \frac{1}{\tau} \int_{t_0 - \tau/2}^{t_0 + \tau/2} \theta(x, y, z = 1/2, t')dt'. \qquad (5)$$

Snapshot data is output periodically and $t_0$ is the time scale for this output interval. Both fields are transformed onto a polar wavevector grid in Fourier space giving $\hat{U}(k, \phi_k; \tau, t_0)$ and $\hat{\Theta}(k, \phi_k; \tau, t_0)$. The variable $k$ stands for the magnitude of the wave vector, and $\phi_k$ is the angle the wavevector makes in wavenumber space. Azimuthally averaged Fourier spectra (see Fig. 4b) are given by

$$E_\omega(k; \tau, t_0) = \frac{1}{2\pi} \int_0^{2\pi} |\hat{\omega}(k, \phi_k; \tau, t_0)|^2 d\phi_k, \qquad (6)$$

with $\hat{\omega} = \{\hat{U}, \hat{\Theta}\}$. The spectrum $E(k)$ is also known as the structure function $S(k)$ and is a useful way to find the wavenumber for a two-dimensional spectrum[36]. All spectra $E_\omega(k; \tau, t_0)$ show a global maximum. An additional average over all $t_0$ yields a unique maximum wavenumber $k^*_{U,\Theta} = 2\pi/\hat{\lambda}_{U,\Theta}$ which depends on Ra and Pr as shown in Fig. 4c, d. The wavelength $\hat{\lambda}_{U,\Theta}(\text{Ra}, \text{Pr})/2$ is the characteristic mean width of the superstructure rolls as sketched in panel a of Fig. 4. We note that the spectra $E_\omega(k; \tau, t_0)$ do not vary significantly with $t_0$, in particular in respect to the maximum wavenumber $k^*$. The characteristic wavelengths in Fig. 4c, d are larger than the critical wavelength $\hat{\lambda}_c = 2\pi/k_c \approx 2$ at the onset of convection with $\text{Ra}_c = 1708$[23]. It is seen that the wavelength grows with Ra at fixed

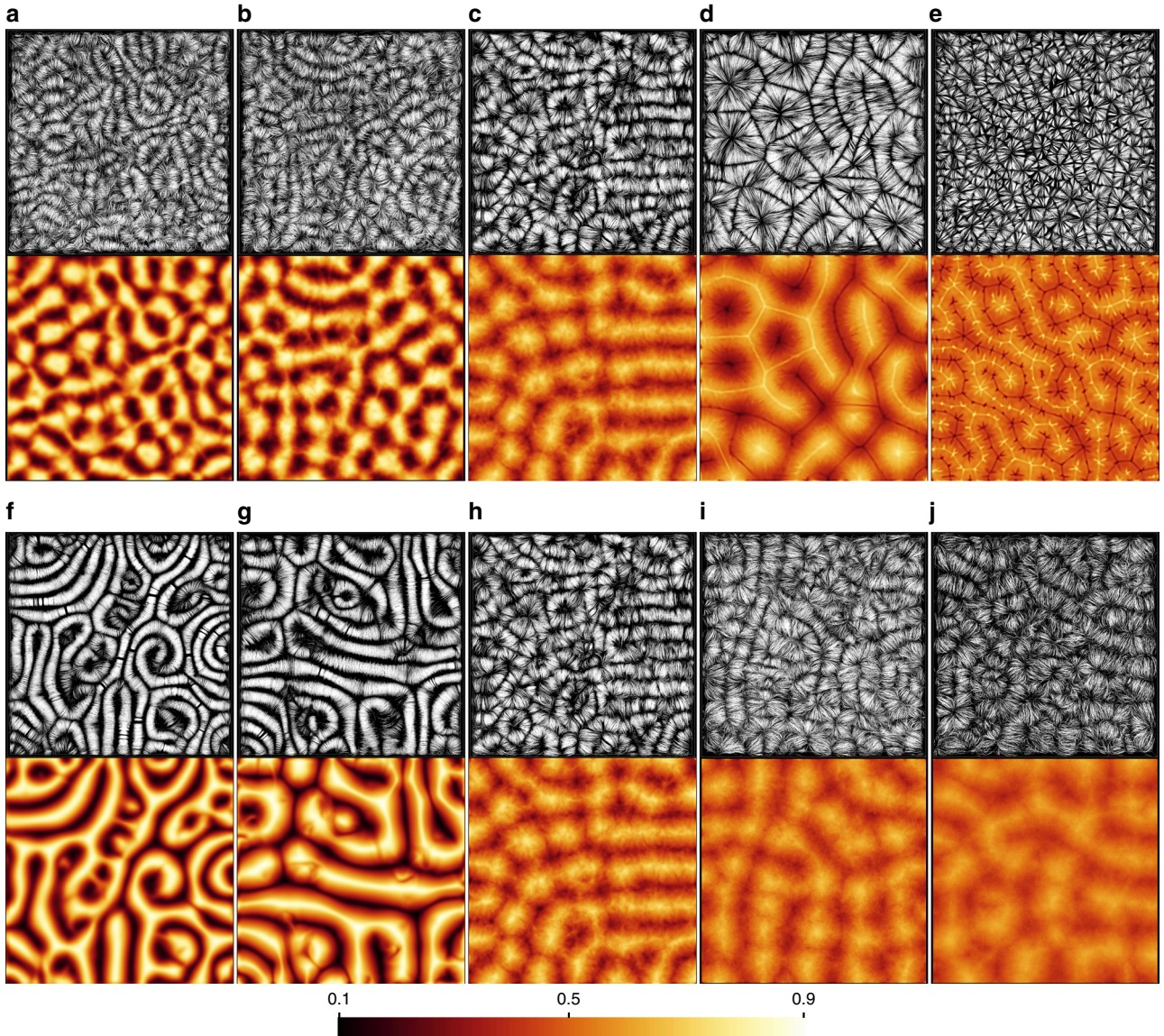

**Fig. 3** Turbulent superstructures at different Rayleigh and Prandtl numbers. For each of the simulations field line plots of the time-averaged velocity (top rows) and the corresponding time-averaged temperature in the midplane (bottom rows) are displayed. Turbulent Rayleigh-Bénard convection at a Rayleigh number of Ra = $10^5$ (**a**) at Pr = 0.005, (**b**) at Pr = 0.021 (see also Fig. 1b, d), (**c**) at Pr = 0.7, (**d**) at Pr = 7, and (**e**) at Pr = 70. Averaging times in **a**–**e** are 21 free-fall times ($T_f$) for Pr = 0.005, 27 $T_f$ for Pr = 0.021, 57 $T_f$ for Pr = 0.7, 207 $T_f$ for Pr = 7, and 375 $T_f$ for Pr = 70. Turbulent Rayleigh-Bénard convection at a Prandtl number of Pr = 0.7 at (**f**) Ra = 5 × $10^3$, (**g**) at Ra = $10^4$, (**h**) at Ra = $10^5$ (same panels as in **c**), (**i**) at Ra = $10^6$, and (**j**) at Ra = $10^7$. The averaging times are now 54 $T_f$ for Ra = 5 × $10^3$, 72 $T_f$ for Ra = $10^4$, 57 $T_f$ for Ra = $10^5$, 66 $T_f$ for Ra = $10^6$, and 72 $T_f$ for Ra = $10^7$. All shown cross sections are 25$H$ × 25$H$ with $H$ being the height of the convection layer

Pr. The dependence of the wavelength on the Prandtl number at fixed Rayleigh number in our data indicates a growth up to Pr ~ 10 and a subsequent decrease for even higher values which is in agreement with ref. [15] for smaller Γ. In Supplementary Fig. 4 and Supplementary Note 3 we demonstrate that nearly the same scales can be obtained by an analysis of the two-point correlation functions in physical space.

Interestingly, Fig. 4 also shows that $\hat{\lambda}_\Theta \gtrsim \hat{\lambda}_U$. At the onset of convection, both wavelengths are exactly the same since both fields are perfectly synchronized in the midplane. Hot fluid is advected upwards ($\theta, u_z > 0$) while cold fluid is brought downwards ($\theta, u_z < 0$). This perfect synchronicity breaks down with increasing Ra since the temperature field is not only advected by vertical velocity component across the midplane, but also by rising horizontal velocity fluctuations. They expand the

temperature patterns compared to those of the vertical velocity component which manifests in a somewhat larger wavelength $\hat{\lambda}_\Theta$. We quantified this effect by the calculation of a horizontal Péclet number, which is given by $Pe_h = v_h H/\kappa$. It relates the (horizontal) advection of the temperature by the velocity field to temperature diffusion. Here, $v_h(z = 1/2) = (\langle u_x^2 \rangle + \langle u_y^2 \rangle)^{1/2}$. The Péclet number is always larger than 10 which underlines a dominance of convection in comparison to diffusion.

With the characteristic width of the superstructure rolls (or cells) of $\hat{\lambda}_U/2$ determined, we can now define the characteristic turnover time for a fluid parcel. We estimate this time scale by an elliptical circumference, $\ell \approx \pi(a + b)$ with $a$ and $b$ (see again Fig. 4a) being the half-axes, and root mean square velocity of the turbulent flow. The characteristic time scale of the turbulent superstructures, beyond which the gradual evolution of

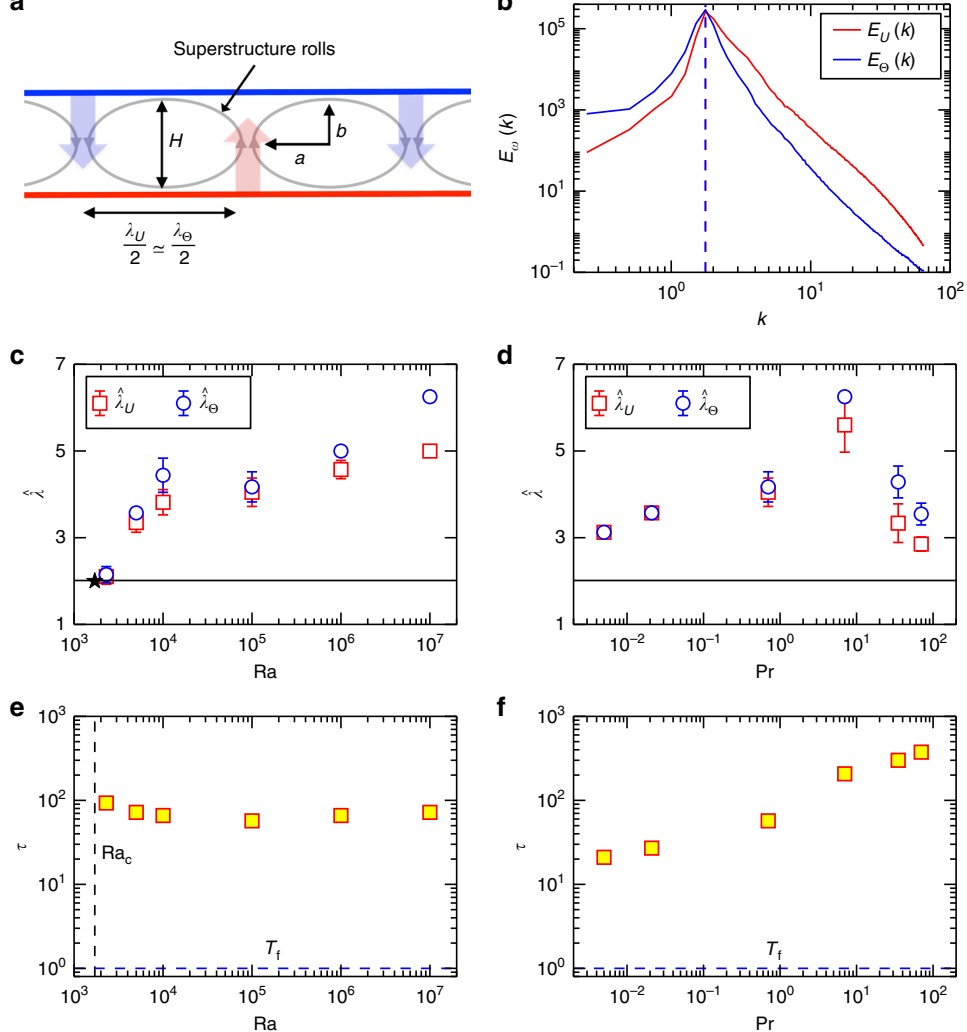

**Fig. 4** Characteristic times and scales of turbulent superstructure patterns. **a** Sketch of the turbulent superstructure with the rolls (or cells) formed by upwelling hot and downwelling cold fluid. The characteristic scale of the superstructures is half the roll pattern wavelength, $\hat{\lambda}_{U,\Theta}$. **b** Azimuthally averaged power spectra, $E_U(k)$ and $E_\Theta(k)$. Data are for $Pr = 0.021$ and $Ra = 10^5$ as an example. The power spectra are determined for each snapshot and then averaged over all snapshots. Maximum wavenumbers of both power spectra are indicated by the vertical dotted line. **c** Rayleigh number dependence of characteristic wavelength $\hat{\lambda}_{U,\Theta}$ of the superstructures for turbulent convection in air. **d** Prandtl number dependence of characteristic wavelength of superstructures at $Ra = 10^5$. All error bars are plotted though some are too small to see. The error bars are given by $\pm \Delta\hat{\lambda}_{U,\Theta} = (2\pi/k_{U,\Theta}^{*2})\Delta k$ with $\Delta k = \max(k_{U,\Theta}^*(t_0)) - \min(k_{U,\Theta}^*(t_0))$. The solid lines in **c**, **d** mark the critical wavelength $\hat{\lambda}_c = 2\pi/k_c$ with $k_c = 3.117$ at the onset of convection[23]. **e** Characteristic time $\tau$ as a function of Rayleigh number at $Pr = 0.7$. **f** Characteristic time as a function of the Prandtl number at $Ra = 10^5$. The free-fall time, $T_f$, is also indicated in both panels as a horizontal dashed line. The vertical dashed line in panel **e** stands for $Ra = Ra_c = 1708$. Characteristic wavelengths are given in units of height $H$, characteristic times in units of $T_f$

the large-scale patterns proceeds, is given by

$$\tau(Ra, Pr) \approx 3 \frac{\ell}{u_{rms}} \approx 3 \frac{\pi\left(\frac{1}{4}\hat{\lambda}_U + \frac{1}{2}H\right)}{\left\langle u_x^2 + u_y^2 + u_z^2 \right\rangle_{V,t}^{1/2}}. \quad (7)$$

Figure 4e, f displays these computed times as a function of Ra and Pr. The prefactor of 3 in Eq. (7) accounts for the fact that an individual fluid parcel is not perfectly circulating around in such a roll when the flow is turbulent. We tested that different prefactors of same order of magnitude do not change the results qualitatively. The characteristic time $\tau$ is found to be nearly unchanged at the fixed Prandtl number. It increases with Pr at

fixed Ra, remaining however always well below the upper bound, the dissipation time scale $T_d$ (see Table 1).

**Radially averaged power spectra.** On time scales larger than $\tau$ the turbulent superstructure patterns are found to evolve by slow changes in orientation and topology. This can be quantified by an angular spectral analysis[37]. We take the radially averaged power spectrum of temperature $\Theta$ which is given by

$$E_\Theta(\phi_k; \tau, t_0) = \frac{1}{k_{max}} \int_0^{k_{max}} \left|\hat{\Theta}(k, \phi_k; \tau, t_0)\right|^2 dk, \quad (8)$$

and plot the spectra in Fig. 5 versus time $\tau$. The wavenumber $k_{max}$ in Eq. (8) denotes a cutoff with $k_{max} \gg k_\Theta^*$. Local maxima in this spectrum indicate now a preferential orientation of parallel rolls. The slow evolution of the turbulent superstructures becomes

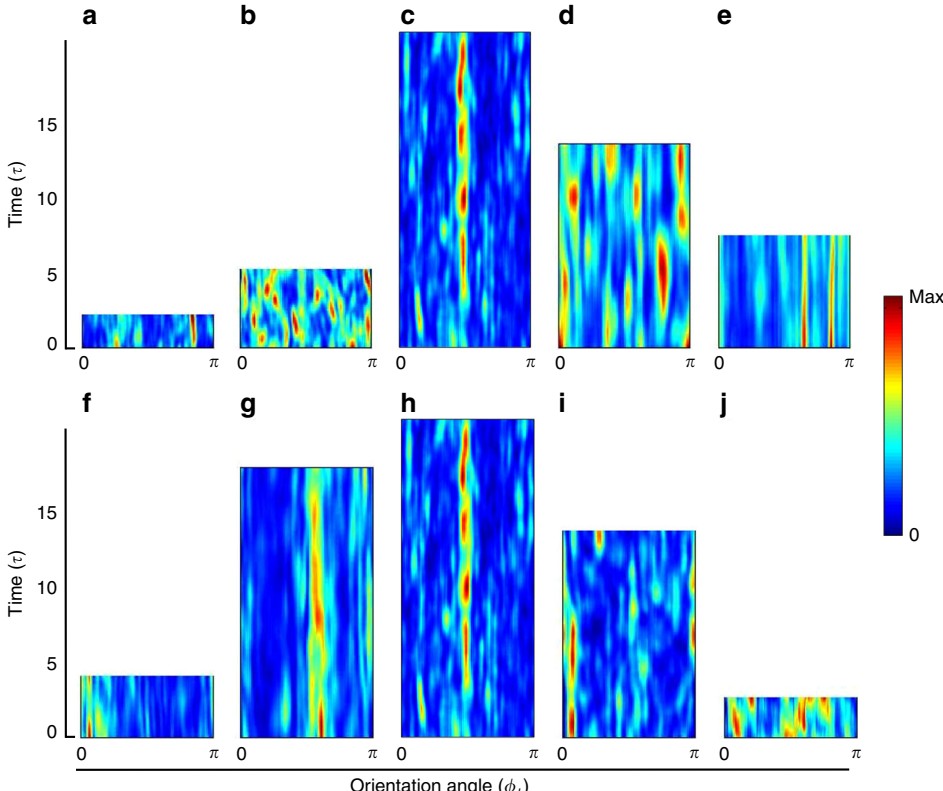

**Fig. 5** Slow time evolution of superstructures. Color density plot of the radially averaged temperature power spectrum $E_\Theta(\phi_k; \tau, t_0)$ as a function of time $t_0$ (in units of $\tau$) and orientation angle $\phi_k$. We plot the angle in wavenumber space, $\phi_k = \arctan(k_y/k_x)$ between 0 and $\pi$. All panels are rescaled with their corresponding characteristic time $\tau$. The top row shows data for Ra $= 10^5$. **a** Pr $= 0.005$, **b** Pr $= 0.021$, **c** Pr $= 0.7$, **d** Pr $= 7$, and **e** Pr $= 70$. The bottom row shows data for Pr $= 0.7$. **f** Ra $= 5 \times 10^3$, **g** Ra $= 10^4$, **h** Ra $= 10^5$, **i** Ra $= 10^6$, and **j** Ra $= 10^7$

visible by the slow variation of the local maxima in the spectrum in all presented runs. We can identify in all cases a small number of local maxima that grow and then decay with time. As the old maxima decay, new ones set in that are shifted by discrete angles from the old ones. This suggests secondary modulations of the dominant roll pattern. We also see that for the highest Pr the maxima persist for a very long period while they switch more rapidly in case of the lower Pr. This behavior is reminiscent of cross-roll or skewed varicose instabilities that have been studied in detail in weakly nonlinear convection above onset[26, 38].

We have measured the duration of the global maxima in all plots in Fig. 5. A mean lifetime $\tau_{spec}$ is obtained by first finding the global maxima of the spectra, $E_\Theta(\phi_k; \tau, t_0)$ at $t_0$. Then the mean lifetime is determined from the duration of these maxima. The angular position is monitored with a resolution of approximately 6 degrees. The results for $\tau_{spec}$ obtained by this alternative method are shown in Fig. 6a, b in units of $\tau$. The mean lifetime is consistently of the order of $\tau$ (see Fig. 4e, f), except for the highest Pr.

**Connection to boundary layers**. Figure 4c, d show that the characteristic scale of the superstructures varies with Pr and Ra. The trends with Prandtl number at fixed Rayleigh number and with Rayleigh number at fixed Prandtl number can be explained with the horizontal Péclet number $Pe_h$ again. We detect a growth of this number from $Pe_h = 7$ to 500 for Ra $= 10^4$ to $10^7$ at Pr $= 0.7$. It is furthermore found that $Pe_h$ grows by a factor of 4 from Pr $= 0.005$ to 0.7 and falls off again by the same factor to Pr $= 70$. The stirring of temperature by the horizontal flow is maximal at

Pr $\sim 1$ which supports the maximum of $\hat{\lambda}_\Theta$ (and thus in turn that of $\hat{\lambda}_U$) in Fig. 4d.

Figure 7 connects the time-averaged temperature field structure in the boundaries at the top and bottom plates with that in the midplane at $z = 1/2$ for two different Prandtl numbers. We display the time-averaged vertical temperature derivative $\langle \partial T/\partial z \rangle_\tau$ at $z = 0, 1$ in Fig. 7a, e, f, j. This field is one way to highlight the most prominent thermal plume ridges[39]. As expected, thinner filaments are observed for a higher Pr while the contours appear somewhat blurred and coarse grained for the lowest Prandtl number.

Figure 7b, d, g, i displays a zoom of the same data together with the field lines of the time-averaged skin friction field $\langle \mathbf{s} \rangle_\tau = (\langle \partial u_x/\partial z \rangle_\tau, \langle \partial u_y/\partial z \rangle_\tau)$ at the plates. The two-dimensional skin friction field is composed of the two non-vanishing components of the velocity gradient tensor at $z = 0, 1$. It contains sources and sinks and is fully determined by its critical points, $\mathbf{s} = 0$[40, 41]. These critical points are either unstable nodes, stable nodes or saddles, and much less frequently unstable and stable foci. Groups of saddles and stable nodes are correlated with local regions of the formation of dominant plumes while unstable nodes are mostly found where colder (hotter) fluid impacts the bottom (top) plate. This is still very clearly visible for the time average at Pr $= 7$ in the bottom row of the figure, but does also hold for the low-Prandtl-number data displayed here. The structures at the top plate display the same plume ridges, but are shifted by a roll-width when compared with those at the bottom plate, as is expected for a system of parallel rolls. Figure 8 shows an even stronger magnification of temperature derivative and skin friction field for instantaneous snapshots also indicating the determined critical points in Fig. 8c, g.

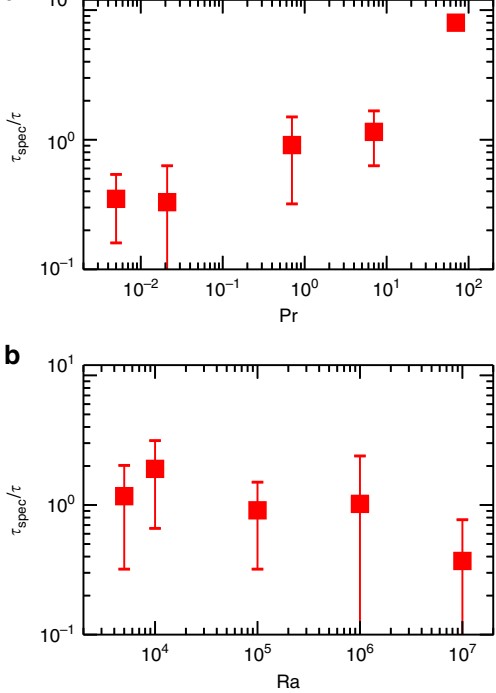

**a**

**b**

**Fig. 6** Mean lifetime of turbulent superstructure $\tau_{spec}$ in units of time scale $\tau$. The lifetime is obtained from the duration of global maxima of the radially averaged temperature spectra. **a** Mean lifetime versus Prandtl number at $Ra = 10^5$. **b** Mean lifetime versus Rayleigh number at $Pr = 0.7$. The error bars correspond to the standard deviation of the lifetimes

Fig. 7c, h shows the turbulent superstructure of temperature in the midplane with local maxima and minima exactly where the hot and cold plume ridges are present at the plates, respectively. These dominant ridges are the ones that persist as the superstructures when the time-averaging over $\tau$ is performed. Figure 7 thus demonstrates that the turbulent superstructures are directly connected to the strongest thermal plumes in the boundary layers. This plume formation process is determined by two aspects: (i) the molecular diffusivity of the temperature field (and the resulting differences in the thicknesses of thermal and viscous boundary layers) and (ii) the typical variation scale of the horizontal velocity field near the walls that forms the plume ridges by temperature field advection. While the first aspect will affect the shape of the plume ridges and thus the characteristic thickness scale of the local temperature maxima and minima in the midplane, the second one is directly connected to the spacing of the dominant temperature structures in the midplane and thus the width of the large-scale circulation rolls and cells that fill the layer. The divergence of the time-averaged skin friction field which is obtained by

$$\text{div } \boldsymbol{s} = \left.\frac{\partial^2 u_x}{\partial x \partial z}\right|_{z=0} + \left.\frac{\partial^2 u_y}{\partial y \partial z}\right|_{z=0} = -\left.\frac{\partial u_z^2}{\partial z^2}\right|_{z=0}, \quad (9)$$

(plus time averaging) can be considered as a blueprint of the alternating impact (source with div $\boldsymbol{s} > 0$) and ridge formation (sink with div $\boldsymbol{s} < 0$) regions. The skin friction field is thus a key to understanding the clustering of thermal plumes near the wall, a phenomenon which has been reported for example in ref. [42]. The same picture holds at the top plate.

Figure 9 underlines this correlation by means of the power spectra of the temperature in the midplane, the vertical temperature derivative at the plates and the divergence of the

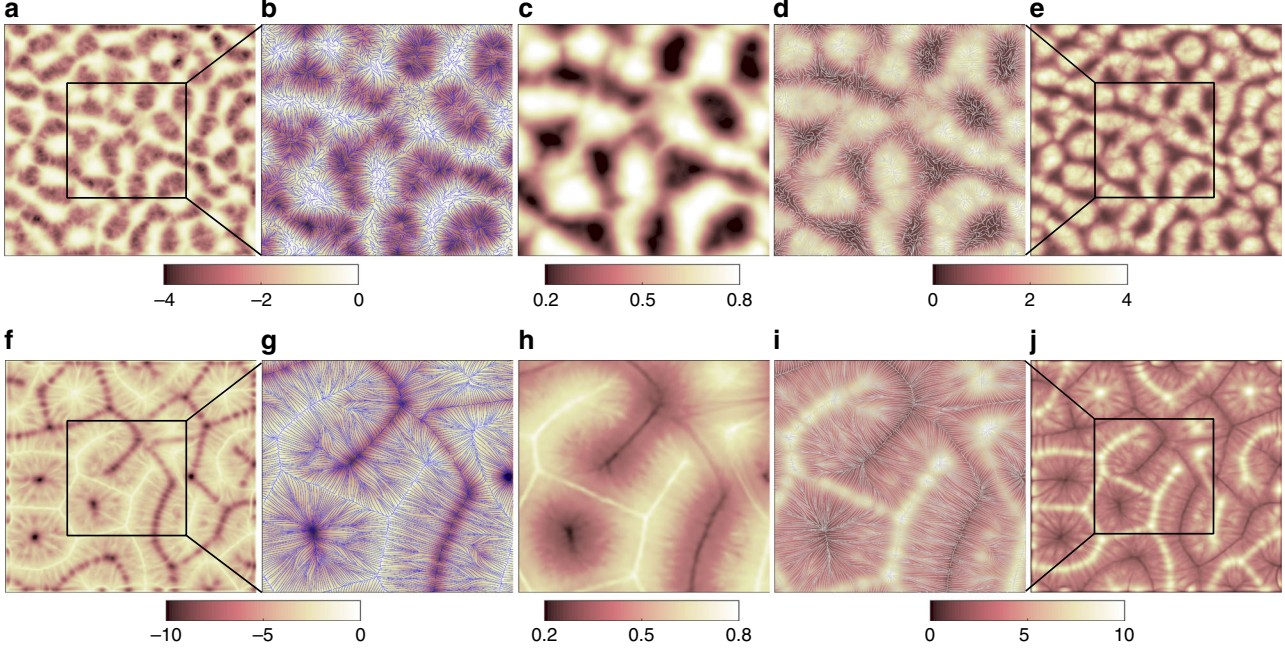

**Fig. 7** Temperature field structure averaged over time $\tau$ in boundary layer and midplane. The data in top row (**a**–**e**) of the figure are obtained for the case of $Pr = 0.005$, the data in bottom row (**f**–**j**) for $Pr = 7$, both at $Ra = 10^5$. **a**, **f** Contours of $\langle \partial T / \partial z \rangle_\tau$ at $z = 0$. **e**, **j** Contours of $-\langle \partial T / \partial z \rangle_\tau$ at $z = 1$. Boxes in these plots indicate the magnification region which is a quarter of the full cross section. **b**, **d** and **g**, **i** are magnifications of **a**, **e** and **f**, **j**, respectively. In addition, we display field lines of the corresponding time-averaged skin friction field $\langle \boldsymbol{s} \rangle_\tau$ in blue (white) for the bottom (top) plate. **c**, **h** show the corresponding time-averaged temperature field $\Theta$ in the midplane at $z = 1/2$. Color bars are added to the corresponding figures. The size of **b**–**d** and **g**–**i** is the same

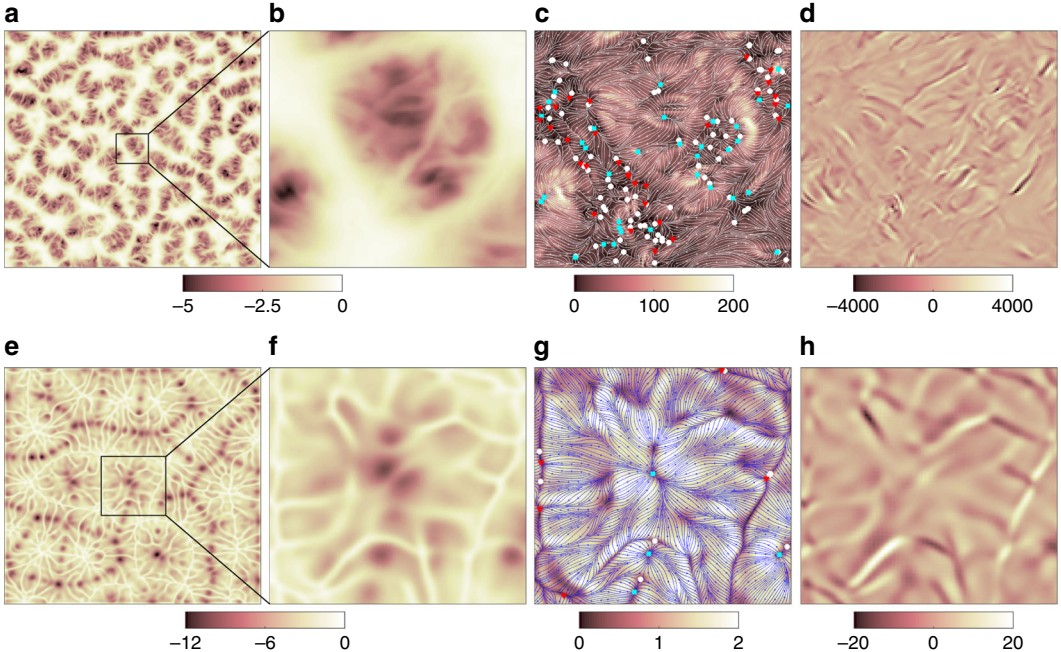

**Fig. 8** Instantaneous structure of the temperature derivative $\partial T/\partial z$ and skin friction field $\boldsymbol{s}$. Results at the bottom plate are displayed in magnification. **a–d** are for Pr = 0.005 and **e–h** for Pr = 7. Both data sets are obtained for Ra = $10^5$. **a, e** show the derivative $\partial T/\partial z$ at the plate $z = 0$. **b, f** replot these data in the magnified section. For **b** $-1.6 \leq x, y \leq 1.6$ and for **f** $-3.1 \leq x, y \leq 3.1$ are taken. **c, g** display the field lines of the skin friction field and the critical points as symbols. White circles correspond to saddles, cyan squares to unstable nodes or foci and red triangles to stable nodes or foci. The colored background stands for the magnitude $s = |\boldsymbol{s}|$. **d, h** show contours of the divergence of the skin friction field. Sizes of **c, d** are the same as **b** and those of **g, h** are the same as **f**

skin friction field. We have applied again the sliding time average over $\tau$. All three spectra are found to peak at the same scale (except Pr ≥ 7 where the scales however are still comparable). Our result is thus robust with respect to Pr and underlines that the same dynamical processes are at work for all Prandtl numbers. As seen in Fig. 9, the characteristic scale of the skin friction divergence is expected to decrease when the Prandtl number gets smaller. This in turn is visually confirmed by Fig. 8d, h where we note that the magnification section for Pr = 0.005 was chosen smaller in order to display features better. It is documented in refs. [29, 30], that the Reynolds number increases significantly when Pr decreases at constant Ra thus indicating a much more vigorous fluid turbulence, both in the bulk and in the boundary layers. Thus the spatially extended advection patches of the horizontal velocity field, as visible in the magnification for the case for Pr = 7 in Fig. 7g, i, will not persist for low-Prandtl-number convection. Finally, we stress that only a strong correlation between boundary layer dynamics and patterns in midplane is shown. Our analysis cannot determine if the observed characteristic scale of the superstructure is a direct result of the boundary layer dynamics.

## Discussion
Our main motivation was to study the large-scale patterns in turbulent convection which are termed turbulent superstructures. We then analyzed the characteristic length and time scales associated with these turbulent superstructures as a function of Rayleigh and Prandtl numbers and found a separation between large-scale, slowly evolving structures and small-scale, rapidly turning vortices and filaments. The system that we have chosen is the simplest setting for a turbulent flow that is initiated by temperature differences, a Rayleigh-Bénard convection flow between uniformly heated and cooled plates. This flow has already been studied intensively with respect to pattern formation in the

weakly nonlinear regime above the onset of convection at Ra = $Ra_c$, as documented in the cited reviews[24–26]. Our study shows that patterns of rolls and cells continue to exist into the fully turbulent and time-dependent flow regime once the small-scale fluctuations of the temperature and velocity fields are removed.

Prandtl numbers that vary here over more than four orders of magnitude change the character of convective turbulence drastically from a highly inertia-dominated Kolmogorov-type turbulence at the lowest Pr to a fine-structured convection at the highest Pr. This results in a strong dependence of the characteristic spatial and temporal separation scales that are necessary to describe the gradual large-scale evolution of the flow at hand. These spatial separation scales are found to continuously increase up to Pr ≲ 10 and to decay for Pr ≳ 10 for the parameter values that we were able to cover here which is in agreement with ref. [15]. A saturation of the characteristic scale might occur for the opposite limit, Pr → 0. Our data indicate such a behavior which is supported by previous studies at zero-Prandtl convection by Thual[7]. There they found only small differences between Pr = 0.025 and the singular limit Pr = 0. However these former studies have been conducted in much smaller boxes at significantly smaller spectral resolutions.

A further interesting observation that was made in the present study is the connection between the mean scales of the turbulent superstructure patterns analyzed in the midplane and those of the near-wall flows. Our analysis suggests that the characteristic scales of large-scale superstructures are correlated with the thermal plume ridges in the boundary layers. We showed for all Pr that the maximum wavenumber of the temperature spectrum in the midplane $k_\Theta^*$ nearly perfectly coincides with the wavenumber at which the power spectrum of the divergence of the skin friction field peaks. The latter wavenumber characterizes the mean distance of impact (div $\boldsymbol{s} > 0$) and ejection (div $\boldsymbol{s} < 0$) regions at the

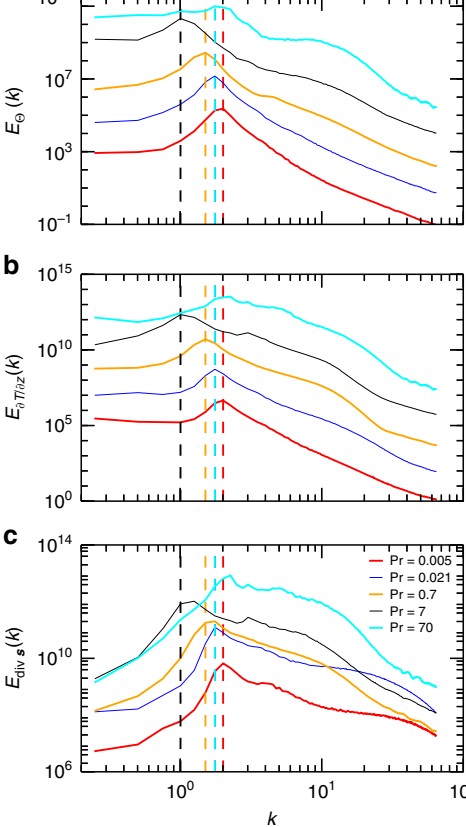

**Fig. 9** Scale correlations of bulk and boundary layer. Time-averaged spectra of temperature Θ in the midplane (**a**), of vertical temperature derivative $\partial T/\partial z$ at the bottom and top plates (**b**), and of skin friction field divergence at the bottom and top plates (**c**) are compared for five different Prandtl numbers. The vertical dashed lines denote the maximum wavenumbers $k_\Theta^*$ and are replotted in **b** and **c**. The dashed lines for $Pr = 0.021$ and 70 (blue and cyan) collapse. Spectra are shifted vertically for better visibility

walls. It is thus the characteristic variation scale of the horizontal velocity field that advects the hot (cold) fluid together at the bottom (top) boundary to form prominent thermal plume ridges. The interplay between the thermal and viscous boundary layers of different thicknesses could thus be responsible for the variation of the characteristic superstructure scale with growing Pr. The viscous boundary layer becomes even thicker as Pr increases and velocity fluctuations decrease, thus generating more coherent advection patterns. Competing boundary layers that control transport and structure formation in convection flows have been discussed in other settings, for example in ref. [43] for rapidly rotating convection.

The characteristic superstructure scales which we have detected in the present work suggest a scale separation for convective turbulence. There is the fast convective motion below the characteristic width of individual circulation rolls or cells on times smaller than several tens of free-fall times. Then after the removal of the small-scale turbulence, the large-scale patterns of rolls are revealed and these fill the whole layer and vary slowly on time scales larger than a few hundreds of free-fall times. The latter dynamic processes can be of interest for a global effective description of mesoscale convection phenomena in atmospheric turbulence[44] or of pattern formation in a scale range between solar granulation and supergranulation[45]. In contrast to rapidly

rotating convection flows or magnetoconvection in the presence of strong external magnetic fields, the present RBC flow permits a mathematically rigorous asymptotic expansion that generates simplified equations for the dynamics of these patterns (see e.g., ref. [46]). The unresolved dynamics at the fine and fast scales below $\hat{\lambda}_{\Theta,U}$ and $\tau$ will be modeled empirically. This is being further investigated and will be reported elsewhere.

## Methods

**Numerical simulations.** We solve the coupled three-dimensional equations of motion for velocity field $u_i$ and temperature field $T$ in the Boussinesq approximation of thermal convection:

$$\frac{\partial u_i}{\partial x_i} = 0, \tag{10}$$

$$\frac{\partial u_i}{\partial t} + u_j \frac{\partial u_i}{\partial x_j} = -\frac{\partial p}{\partial x_i} + \sqrt{\frac{Pr}{Ra}} \frac{\partial^2 u_i}{\partial x_j^2} + T\delta_{i3}, \tag{11}$$

$$\frac{\partial T}{\partial t} + u_j \frac{\partial T}{\partial x_j} = \frac{1}{\sqrt{RaPr}} \frac{\partial^2 T}{\partial x_j^2}, \tag{12}$$

with Rayleigh number $Ra = g\alpha\Delta T H^3/(\nu\kappa)$ and Prandtl number $Pr = \nu/\kappa$. The indices $i, j$ can have the values $x, y, z$. The equations are made dimensionless by cell height $H$, free-fall velocity $U_f = \sqrt{g\alpha\Delta T H}$ and the imposed temperature difference $\Delta T$ between the bottom and top plates. The aspect ratio $\Gamma = L/H = 25$ with the cell length $L$. The variable $g$ stands for the acceleration due to gravity, $\alpha$ is the thermal expansion coefficient, $\nu$ is the kinematic viscosity, and $\kappa$ is the thermal diffusivity. No-slip boundary conditions for the fluid are applied at all walls. The sidewalls are thermally insulated and the top and bottom plates are held at constant dimensionless temperatures $T = 0$ and 1, respectively. For a comparison with periodic boundary conditions at the sidewalls we refer to Supplementary Fig. 5, Supplementary Table 1 and Supplementary Note 4. The equations are numerically solved by the Nek5000 spectral element method package (https://nek5000.mcs.anl.gov).

The turbulent heat and momentum transfer is quantified by the Nusselt Nu and Reynolds number Re, respectively. They are defined in dimensionless form as

$$Nu = 1 + \sqrt{RaPr}\langle u_z T\rangle_{V,t}, \tag{13}$$

$$Re = \sqrt{\left\langle u_x^2 + u_y^2 + u_z^2\right\rangle_{V,t} \frac{Ra}{Pr}}, \tag{14}$$

with $\langle\cdot\rangle_{V,t}$ being a full volume–time average. We have verified that the spectral resolution is sufficient in correspondence with criteria that are summarized in ref. [47]. We compared two pairs of runs at different spectral resolution and tested that the results for the global transport of heat and momentum across the layer are the same. These are runs 3 and 3a at $Pr = 0.7$ and $Ra = 10^5$ as well as runs 4 and 4a at $Pr = 7$ and $Ra = 10^5$ (see Table 1). We also verified that the vertical profiles of the plane- and time-averaged kinetic energy and thermal dissipation rates exhibit smooth curves across the element boundaries[47].

**Data availability**. Data that support the findings of this study are available from the corresponding author on request.

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

## Acknowledgements

A.P. and J.D.S. acknowledge support by the Deutsche Forschungsgemeinschaft within the Priority Programme Turbulent Superstructures under Grant no. SPP 1881. We acknowledge supercomputing time at the Blue Gene/Q JUQUEEN at the Jülich Supercomputing Centre by large-scale project HIL12 of the John von Neumann Institute for Computing and at the SuperMUC Cluster at the Leibniz Supercomputing Centre Garching by large-scale project pr62se. We thank Eberhard Bodenschatz, Bruno Eckhardt, Keith Julien, Raymond A. Shaw, and Katepalli R. Sreenivasan for helpful comments and discussions. We acknowledge support for the Article Processing Charge by the Thuringian Ministry for Economic Affairs, Science and Digital Society and the Open Access Publication Fund of the Technische Universität Ilmenau.

## Author contributions

All the three authors made significant contributions to this work. All authors designed the numerical experiments and analyzed the data. A.P. and J.S. ran the production simulations at the supercomputing sites in Garching and Jülich. All authors discussed the results and wrote the paper together.

## Additional information

**Competing interests:** The authors declare no competing interests.

