## [Peer Review File · Nature Communications]

Reviewer #1 (Remarks to the Author):

The authors present results of direct numerical simulations (DNS) of turbulent Rayleigh-Bénard convection (RBC) in a square cell with large aspect ratio. Simulations were performed for a moderately high Rayleigh (Ra) number of 10^5 and Prandtl (Pr) numbers ranging from 0.005 to 70 on the one hand and for $Pr = 0.7$ and Ra numbers ranging from 5×10^3 to 10^7 on the other hand. For each case small- and large-scale structures are separated by time-averaging of instantaneous flow fields. The found large-scale flow structures are denoted as superstructures.

The paper is well written and contains interesting results regarding RBC in a large aspect ratio cell for a wide range of Pr numbers. The main message is that the time-averaged flow is organized in large-scale flow patterns which are similar to those found at much lower Rayleigh numbers. It is at least questionable to denote these large-scale motions as “superstructures”. Further, it is well-known from studies of turbulent RBC in air ($Pr = 0.7$) and other common fluids that time-averaged large Ra number flows are organized in large-scale patterns. However, the present paper confirms the latter observation for a wide range of Pr numbers.

Therefore, the paper is recommended for publication.

Reviewer #2 (Remarks to the Author):

Review of Pandey et al “Turbulent superstructures in Rayleigh-Benard convection”

For a paper to be of broad interest it needs a compelling scientific question, something that makes the reported work challenging, and relevancy to a widely appreciated problem. This paper has all three. The intriguing idea can be simply stated as follows: when a fluid is heated from below by just the right amount, convection begins, in the form of smooth, laminar flow patterns. The variety of patterns that can be achieved is vast, and considerable work has been done to understand what causes the patterns and how they depend on controlling parameters such as the relative values of thermal diffusion and viscosity (as quantified through the Prandtl number). At some point, as the heating is increased, the laminar flow breaks down, resulting in highly random turbulence. The surprise is that, in a sufficiently large container (wide compared to the depth), even the seemingly featureless turbulent flow self-organizes into large-scale, slowly-varying flow patterns. At a qualitative level, anyone who has looked at the cloudy sky and seen hexagonal cells or parallel cloud ‘streets’ has observed a macroscopic manifestation of this phenomenon. And yet, while some empirical rules of thumb have been developed for classifying what patterns can be expected to occur, a rigorous and quantitative description is lacking. This paper addresses that question, and although mysteries still remain, some important clues are revealed. The second needed aspect is something challenging about the work, and here that aspect is clear: the simulations are a computational tour de force, spanning orders of magnitude in Prandtl and Rayleigh numbers, for high-aspect-ratio geometry, and for very long simulation times compared to the turnover time. Simply put, the authors are showing results that are the first of their kind at this level of extensive coverage. Finally, the work is clearly motivated by a host of practical problems, not the least of

which is the hope of someday developing quantitative, physically-based models of atmospheric convection and cloud formation in the earth's turbulent boundary layer. This would have important implications for our ability to forecast weather and properly model earth's climate.

The paper is clearly written, fully describes the computational and analysis tools, and provides both a vast new look at the parameter space of turbulent Rayleigh-Benard convection, as well as important new insights into the organization of turbulent superstructures. I recommend the work be accepted for publication and am confident it will be of interest to the broad audience targeted by Nature Communications. The following points should be considered:

Page 1, bottom of left column: Be clearer about "long-term investigations..." I believe what is meant by long-term is that the simulations run for times very large compared to typical turnover times.

Page 2, left column: The sentence "containing "... correlated with the most prominent ridges in the vertical temperature field derivative at the bottom and top plate..." is somewhat unclear. I believe it would be clearer as "... correlated with the most prominent ridges in the derivative (at the bottom and top plates) of the vertical temperature field..."

Page 2, right column: It is stated that three cases are non-turbulent (panels e, f, and g in Figure 3). What metric is used to determine whether the flow is turbulent? Could it be that the peak and subsequent falloff in Figure 4, panel d for the two largest Pr are a result of those two being non-turbulent, or only weakly turbulent?

Page 3, left column: For the general reader, make it clear that the free-fall time is not the same as the circulation time.

Page 3, left column: Please provide a citation or a brief argument in support of the Γ^2 scaling for averaging time.

Page 4, equation 6: Wavenumbers k and k_ϕ should be more clearly defined here.

Page 5, figure 4: Please comment on the interpretation of an azimuthal average for linear or banded structures, as opposed to cellular structures. Some physical interpretation would help here.

Page 5, right column: A brief definition of Peclet number would be useful for general readers.

Page 5, figure 4: Make clear here, in the caption or in the axis labels, that λ is in units of H , and τ is in units of free-fall time (it is stated in the appendix, but it is significant here in order to understand the interpretation of the numbers).

Page 7, figure 6 caption: "magnification to" should be "magnification of"?

Pages 7 and 8, including figure 6: There is some ambiguity regarding causality here. Perhaps I misunderstand the meaning, but is it the authors' view that the boundary layer features should be considered sources or drivers for the coherent convection patterns, or simply that they are

correlated with those patterns? This subtlety is important to clarify because it has implications for modeling of the large-scale structures.

Page 8, right column: some speculation about why spatial separation scales increase up to $Pr \sim 10$ and then decay beyond that should be included.

Page 9, methods: How would the authors expect results to change without rigid sidewall boundary conditions? Would periodic boundary conditions, such as might be more realistic for an atmospheric boundary layer lead to significant changes?

Reviewer #3 (Remarks to the Author):

The manuscript "Turbulent superstructures in Rayleigh-Bénard convection" has a very promising title because of the notion of superstructure that hitherto had no relation with convection. Unfortunately, the manuscript does not deliver on its promise.

The so called superstructures turn out to be no different from convection cells as we know them since the beginning of the study of Rayleigh-Bénard convection a long time ago. The reference list contains several papers that identified cells or superstructures in turbulent flows in the past. It is not clear from the text what is new apart from the more extended data base obtained from new simulations. For example, the data analysis around equations (4) and (5) does not seem to be exactly the same as in reference [14], but is there a noteworthy improvement?

The paper goes on to discuss relevant time scales of the superstructures. The authors choose to study k_ϕ , which is not well defined when it first appears. If it is the azimuthal angle, ϕ_k would be a more suggestive name. At any rate, this angle is a basically random result for polygonal patterns, and because it is an average, it does not detect the characteristic time of single superstructures. Fig. 5 makes more sense if its only for roll patterns.

I could not understand the mechanism the beginning of the first paragraph on p. 7 (Connection to boundary layers) attempts to explain, especially if "erratic variations of temperature filaments" are supposed to explain variation in superstructure size as large as observed. The connection between cell walls and plumes again is not really new. Cell walls are almost by definition places where most plumes fall or rise, either because they entrain the mean flow, or because the roll flow detaches the plumes (cause and effect presumably cannot be separated). I have also seen visualizations of the type of fig. 6 before, at least at high Pr .

In summary, I do not see in this manuscript the kind of novelty I would expect in a high profile publication. It is true that the authors have pushed their simulations to lower Prandtl numbers and larger box sizes than anyone before, but they fail to extract from their data some new physical understanding. Even though the quantitative results will be of interest to an expert audience, I do not think they belong into Nature Communications.

Response to Reviewer #1

First, we wish to thank the Reviewer for her/his careful reading and the constructive comments. The resulting (and other) changes have been highlighted in color in the revised manuscript PDF file.

The authors present results of direct numerical simulations (DNS) of turbulent Rayleigh-Bénard convection (RBC) in an square cell with large aspect ratio. Simulations were performed for a moderately high Rayleigh (Ra) number of 10^5 and Prandtl (Pr) numbers ranging from 0.005 to 70 on the one hand and for $Pr=0.7$ and Ra numbers ranging from 5×10^3 to 10^7 on the other hand. For each case small- and large-scale structures are separated by time-averaging of instantaneous flow fields. The found large-scale flow structures are denoted as superstructures.

The paper is well written and contains interesting results regarding RBC in a large aspect ratio cell for a wide range of Pr numbers. The main message is that the time-averaged flow is organized in large-scale flow patterns which are similar to those found at much lower Rayleigh numbers. It is at least questionable to denote these large-scale motions as ‘superstructures’. Further, it is well-known from studies of turbulent RBC in air ($Pr=0.7$) and other common fluids that time-averaged large Ra number flows are organized in large-scale patterns. However, the present paper confirms the latter observation for a wide range of Pr numbers.

Therefore, the paper is recommended for publication.

Answer: We provide a few explanatory remarks here. The standard notion of turbulent superstructures comes from wall-bounded flows where turbulent superstructures (or very large-scale motion) are spatially extended patterns in the velocity fluctuations. But it has not been clear if this notion applies to other flows. This paper demonstrates that they exist in turbulent convection flows for a variety of conditions and also that they have their origin in a linear instability. We have demonstrated that the turbulent flow in a horizontally extended domain is organized into prominent long-lived patterns that exceed the typical scale height of the problem, i.e., the height of the convection layer. We have added new sentences on page 1 (left column).

Indeed, as we state on the first page of our manuscript, one focus of the present work is on a wide range of Prandtl numbers, in particular to very low

Prandtl numbers. In the latter case, the fluid turbulence is highly inertial, even at the moderate Rayleigh numbers discussed here. Even in these cases we find these regular patterns. One further aspect that has not studied to the best of our knowledge so far in this specific context is the slow evolution of these large-scale patterns in *turbulent* convection flows. We have emphasized this point better on page 2 (left column).

Yours sincerely, the authors.

Response to Reviewer #2

First, we wish to thank the Reviewer for her/his careful reading and the comments. In the following we address all of them point by point. The resulting (and other) changes have been highlighted in color in the revised manuscript PDF file.

For a paper to be of broad interest it needs a compelling scientific question, something that makes the reported work challenging, and relevancy to a widely appreciated problem. This paper has all three. The intriguing idea can be simply stated as follows: when a fluid is heated from below by just the right amount, convection begins, in the form of smooth, laminar flow patterns. The variety of patterns that can be achieved is vast, and considerable work has been done to understand what causes the patterns and how they depend on controlling parameters such as the relative values of thermal diffusion and viscosity (as quantified through the Prandtl number). At some point, as the heating is increased, the laminar flow breaks down, resulting in highly random turbulence. The surprise is that, in a sufficiently large container (wide compared to the depth), even the seemingly featureless turbulent flow self-organizes into large-scale, slowly-varying flow patterns. At a qualitative level, anyone who has looked at the cloudy sky and seen hexagonal cells or parallel cloud ‘streets’ has observed a macroscopic manifestation of this phenomenon. And yet, while some empirical rules of thumb have been developed for classifying what patterns can be expected to occur, a rigorous and quantitative description is lacking. This paper addresses that question, and although mysteries still remain, some important clues are revealed. The second needed aspect is something challenging about the work, and here that aspect is clear: the simulations are a computational tour de force, spanning orders of magnitude in Prandtl and Rayleigh numbers, for high-aspect-ratio geometry, and for very long simulation times compared to the turnover time. Simply put, the authors are showing results that are the first of their kind at this level of extensive coverage. Finally, the work is clearly motivated by a host of practical problems, not the least of which is the hope of someday developing quantitative, physically-based models of atmospheric convection and cloud formation in the earth’s turbulent boundary layer. This would have important implications for our ability to forecast weather and properly model earth’s climate.

The paper is clearly written, fully describes the computational and analysis tools, and provides both a vast new look at the parameter space of turbulent Rayleigh-Benard convection, as well as important new insights into the orga-

nization of turbulent superstructures. I recommend the work be accepted for publication and am confident it will be of interest to the broad audience targeted by Nature Communications. The following points should be considered:

Answer: Thank you for these supportive comments.

Page 1, bottom of left column: Be clearer about ‘long-term investigations’ I believe what is meant by long-term is that the simulations run for times very large compared to typical turnover times.

Answer: Times are measured in free fall time units T_f which are given by the layer height H divided by free fall velocity U_f , a velocity which is composed by a set of system parameters (g , α , ΔT , and H). The turnover time scale is the time it takes for a fluid parcel on average to circulate in a large-scale roll that extends between both plates. The turnover time scale depends strongly on the Prandtl number (see Table I in the supplementary material) while T_f is the same time for all simulation cases. The turnover time is about $7 T_f$ for the lowest Pr and more than $100 T_f$ for the highest Pr . Long-term integrations imply always that at least of the order of 10 turnover times have been studied. We have added this information on page 1.

Page 2, left column: The sentence containing “... correlated with the most prominent ridges in the vertical temperature field derivative at the bottom and top plate...” is somewhat unclear. I believe it would be clearer as “... correlated with the most prominent ridges in the derivative (at the bottom and top plates) of the vertical temperature field...”

Answer: Done. We have clarified this sentence on page 2 in the left column.

Page 2, right column: It is stated that three cases are non-turbulent (panels e, f, and g in Figure 3). What metric is used to determine whether the flow is turbulent? Could it be that the peak and subsequent falloff in Figure 4, panel d for the two largest Pr are a result of those two being non-turbulent, or only weakly turbulent?

Answer: We have restated our point more precisely. Certainly, the flows for $Pr = 0.7$, $Ra = 5 \times 10^3, 10^4$ (f,g) and $Pr = 70$, $Ra = 10^5$ (e) are also chaotic and time-dependent. As seen in Fig. 2, they all have a non-vanishing level of temperature and velocity fluctuations about the time-averaged patterns. When the magnitude of the velocity fluctuations exceeds the value of the mean velocity, we call the convection flow fully turbulent, otherwise it is weakly turbulent. This point is stated in the text on page 2. In case (e), we have additionally analysed time-series of the Nusselt number averaged at the heating and cooling plates. These time series show oscillations with respect to time that indicate oscillatory

instabilities and the magnitude of these oscillations is also time-dependent. For the second part of your question, we refer to the last-but-one answer in this response below.

Page 3, left column: For the general reader, make it clear that the free-fall time is not the same as the circulation time.

Answer: Done on page 3.

Page 3, left column: Please provide a citation or a brief argument in support of the Γ^2 scaling for averaging time.

Answer: From the point of view of a dimensional analysis, the horizontal diffusion or dissipation time scale is the largest possible time scale in the system, obtained by substituting the height H by $L = \Gamma H$. Two works from 1985 by Günter Ahlers et al. and Heutmaker et al. (published in Phys. Rev. Lett. side by side) studied the time evolution of thermal convection close to the onset for such very long times in large-aspect ratio cells and detected a gradual evolution of the patterns on this scale. We have inserted these references on page 3 and extended the sentence in this respect. We expect (although we cannot prove) that this time scale is not relevant for a turbulent convection flow. Over multiples of this time scale, the patterns should have evolved such that even the coherence of the superstructures is lost.

Page 4, equation 6: Wave numbers k and k_ϕ should be more clearly defined here.

Answer: We have changed the notation to k and ϕ_k to make it clearer that the variable k stands for the magnitude of the wave vector, and ϕ_k is the angle the wavevector makes in wavenumber space. We have also added this discussion on page 4. Figure 5 is updated accordingly.

Page 5, figure 4: Please comment on the interpretation of an azimuthal average for linear or banded structures, as opposed to cellular structures. Some physical interpretation would help here.

Answer: Since these horizontal slices exhibit two-dimensional patterns, the Fourier transform is two-dimensional, giving either (k_x, k_y) or (k, ϕ_k) depending on your coordinate system. Since the structures are not always exactly aligned with the x and y axes in these cases (see figure 3), one sees in Fourier space a ring with a few prominent peaks (which can also be seen in the angle-time plots in figure 5). Hence the most convenient way to find the dominant wavenumber is through an azimuthal average, as then the exact orientation of the rolls does not need to be known. See Morris et al., Phys. Rev. Lett. **71**, p. 2026-2029 (1993) for more explanation. The azimuthally averaged power spectrum, $E(k)$,

is also known as the structure function $S(k)$ in other literature. We added the Morris reference (now ref. 35) and more explanation on page 4 close to Eq. (6).

Page 5, right column: A brief definition of Péclet number would be useful for general readers.

Answer: The Péclet number Pe compares the advection and diffusion terms in a scalar transport equation (here for temperature). Its definition is similar to that of a Reynolds number Re . The kinematic viscosity ν in Re is replaced by the thermal diffusivity κ in Pe . We have extended the text on page 6 to make this point clear.

Page 5, figure 4: Make clear here, in the caption or in the axis labels, that λ is in units of H , and τ is in units of free-fall time (it is stated in the appendix, but it is significant here in order to understand the interpretation of the numbers).

Answer: Done.

Page 7, figure 6 caption: “magnification to” should be “magnification of”?

Answer: Corrected.

Pages 7 and 8, including figure 6: There is some ambiguity regarding causality here. Perhaps I misunderstand the meaning, but is it the author’s view that the boundary layer features should be considered sources or drivers for the coherent convection patterns, or simply that they are correlated with those patterns? This subtlety is important to clarify because it has implications for modeling of the large-scale structures.

Answer: Yes, the only thing that we can say is that the boundary layer features are correlated with those patterns. Two possibilities exist: either there exist a global instability that sets the characteristic pattern scale and the resulting circulation rolls determine the features in the boundary layer, or the boundary layer dynamics, i.e. the local features close to the wall set a scale by plume clustering that determines the characteristic patterns which we analyse in the midplane. We have added this point on page 7.

Page 8, right column: some speculation about why spatial separation scales increase up to Pr 10 and then decay beyond that should be included.

Answer: One possible explanation builds on similar arguments to those that we used in the text to explain why $\hat{\lambda}_\Theta > \hat{\lambda}_U$. When the Prandtl number is increased at a fixed Rayleigh number, the ratio of the thermal to the viscous boundary layer thickness decreases. It is approximately given by $\delta_T/\delta_v \simeq \sqrt{Re}/(2Nu)$ and

Figure 1: Péclet number Pe_h based on the root mean square velocity of the horizontal components u_x and u_y as a function of Pr . Data for $Ra = 10^5$ (red) and $Ra = 10^6$ (blue) are displayed.

the values are 1.1, 0.4, 0.2 and 0.1 for $Pr = 0.7, 7, 35, 70$ at $Ra = 10^5$ (see Table 1 of supplementary material). The thermal boundary layer is thus increasingly thinner and more deeply embedded inside the viscous boundary layer as Pr grows. As a consequence the thickness of rising and falling thermal plumes (which is comparable to the thermal boundary layer thickness δ_T) decreases such that plumes can less efficiently stir the fluid. This becomes visible in the Péclet number Pe_h which is based on the root mean square of horizontal velocity fluctuations in the center of the layer $v_h = (\langle u_x^2 \rangle + \langle u_y^2 \rangle)^{1/2}$. As figure 1 of this response shows, the Péclet number peaks at $Pr \sim 1$ and decreases for larger Pr . This could explain the decrease of the characteristic scales for larger Pr . We have also analysed this behaviour for $Ra = 10^6$ and $Pr = 0.7, 7, 35, 70$. Again, the wavelengths $\hat{\lambda}_U$ and $\hat{\lambda}_\Theta$ peak for the run at $Pr = 7$ and we find a similar trend for Pe_h (see figure 1). We have added a corresponding paragraph on page 6 (left column).

Page 9, methods: How would the authors expect results to change without rigid sidewall boundary conditions? Would periodic boundary conditions, such as might be more realistic for an atmospheric boundary layer lead to significant changes?

Answer: No-slip walls are taken since we want to compare our simulations directly with laboratory experiments of convection in air and water in the near future (in the same geometry). We have added a paragraph to the supplement that shows a nearly perfect agreement of the statistical quantities. Except for wavelength in Fourier space $\hat{\lambda}_U$, all reported quantities agree in a direct comparison at a smaller aspect ratio of $\Gamma = 16$.

Yours sincerely, the authors.

Response to Reviewer #3

First, we acknowledge the critical comments by the referee which helped us to highlight the new aspects of our work better. In the following we address the comments point by point and hope that she/he changes her/his point of view. The resulting (and other) changes have been highlighted in color in the revised manuscript PDF file.

The manuscript "Turbulent superstructures in Rayleigh-Bénard convection" has a very promising title because of the notion of superstructure that hitherto had no relation with convection. Unfortunately, the manuscript does not deliver on its promise.

Answer: We think that the present work demonstrates for the first time the existence of these large-scale patterns of the temperature and velocity for a broad range of Prandtl numbers that spans more than four orders of magnitude with a particular emphasis on very low Prandtl numbers. In the latter case, the fluid turbulence is highly inertial and it cannot necessarily be expected that the large-scale patterns persist into this parameter range. It is thus the first comprehensive study in this respect that extends the analysis of the spatial and temporal scales of turbulent superstructures from wall-bounded shear flows (see e.g. added new ref. 5) to turbulent convection flows.

The so called superstructures turn out to be no different from convection cells as we know them since the beginning of the study of Rayleigh-Bénard convection a long time ago. The reference list contains several papers that identified cells or superstructures in turbulent flows in the past. It is not clear from the text what is new apart from the more extended data base obtained from new simulations. For example, the data analysis around equations (4) and (5) does not seem to be exactly the same as in reference [14], but is there a noteworthy improvement?

Answer:

On page 1, we clarified our definition of a turbulent superstructure in RBC, which we consider to accurately represent the patterns we have found here when taking the appropriate time average (i.e., over τ as defined in equation 7). We have also now explained that we are extending the idea from wall-bounded shear flows.

Also, novel aspects of the present work are to our view the following ones:

1. Parameter range: We extended the parameter space for turbulent convection flows significantly, both, in terms of the aspect ratio, and more importantly in terms of Prandtl number as stated above. In this regime, no pattern formation studies in a *turbulent* convection flow exist.
2. Scale separation: We extract characteristic spatial and temporal scales that suggest a scale separation into large-scale slowly evolving patterns and rapid small-scale fluid motions. Spatial scales are extracted in multiple ways (Fourier space as in [14] and physical space) leading to consistent results. We provide an argument for the maximum of the characteristic scale as a function of Pr in figure 4d which is based on the horizontal Péclet number.
3. Slow time evolution: We demonstrate the slow temporal evolution of the turbulent superstructures by windowed averaging. Radially averaged spectra provide an alternative way to determine the lifetime of superstructures.
4. Correlation to plume formation: We connect the characteristic scale of the patterns, as observed in the bulk, to the correlations of skin friction field which is analysed at the plates and considered as a blueprint of the near-wall velocity field. We have repeated this analysis for the time-averaged fields now which supports our argumentation even better and updated figure 6 and the corresponding text.

The paper goes on to discuss relevant time scales of the superstructures. The authors choose to study k_ϕ , which is not well defined when it first appears. If it is the azimuthal angle, ϕ_k would be a more suggestive name. At any rate, this angle is a basically random result for polygonal patterns, and because it is an average, it does not detect the characteristic time of single superstructures. Fig. 5 makes more sense if its only for roll patterns.

Answer: We have changed the notation to k and ϕ_k to make it clearer that the variable k stands for the magnitude of the wave vector, and ϕ_k is the angle the wavevector makes in wavenumber space. We have also added this discussion on page 4.

We are not interested in the exact angle, we only want to determine any dominant orientations and how they evolve with time. An angle-time plot, in conjunction with figure 4b, which gives the dominant wavenumber, can provide a lot of information about the patterns, in many cases, and not just for parallel rolls. Many lattice structures can be represented as superpositions of rolls at different orientations (see for example, figure 4.6 of Cross and Hohenberg “Pattern Formation and Dynamics in Non-Equilibrium Physics”, Cambridge, 2009). We found it to be very interesting that the angle-time plots are not more homogeneous, and show, for some cases, very dominant orientations that grow, peak and fade with time, reminiscent of the patterns seen in Cross, Meiron, and Tu, *Chaos* 4, 607 (1994). We have added the additional reference 37. We

added a paragraph to the supplement that now describes the identification of the duration of various peaks in figure 5 with a lifetime of our superstructures. This provides a second alternative way to determine τ .

I could not understand the mechanism the beginning of the first paragraph on p. 7 (Connection to boundary layers) attempts to explain, especially if "erratic variations of temperature filaments" are supposed to explain variation in superstructure size as large as observed. The connection between cell walls and plumes again is not really new. Cell walls are almost by definition places where most plumes fall or rise, either because they entrain the mean flow, or because the roll flow detaches the plumes (cause and effect presumably cannot be separated). I have also seen visualizations of the type of fig. 6 before, at least at high Pr.

Answer: The connection between cell walls and plumes might have been discussed somewhere else. Our flows are turbulent which causes a dispersion of most hot (cold) plumes when they rise (fall) into the bulk. What is shown in Fig. 6 is that the most prominent plume ridges can be correlated with the cell patterns in the midplane. We now show this analysis for the time-averaged fields which supports our argumentation better and have updated the corresponding text on page 7. Furthermore, we include for the first time the structure of the skin friction field into this analysis. The clear connection of this pattern formation to the sources and sinks of the 2d skin friction field is done for the first time and provides, in our view, a better physical understanding of the correlation between boundary layer dynamics and bulk processes.

In summary, I do not see in this manuscript the kind of novelty I would expect in a high profile publication. It is true that the authors have pushed their simulations to lower Prandtl numbers and larger box sizes than anyone before, but they fail to extract from their data some new physical understanding. Even though the quantitative results will be of interest to an expert audience, I do not think they belong into Nature Communications.

Answer: We are pleased with the referee's acknowledgment that we have pushed out simulations to lower Prandtl numbers than anyone before and larger box sizes, but wish to emphasize that the paper is significantly more than just that. Although the reviewer is correct that large-scale structures in turbulent RBC have been found for specific cases before (as in reference 14, for example), there is no general understanding in the RBC community that these structures are present for such a wide range of parameters and that this is a general feature of *turbulent* RBC. In this study, where we cover a wide range of parameters, we have definitively demonstrated the existence of superstructures in turbulent RBC. We have also provided an analysis of the characteristic times and length scales of these structures as a function of Rayleigh and Prandtl number. We think this paper makes important enough contributions to be published in Nature Communications so that its general lessons can be disseminated to a wider

community: in short, we think that the results demonstrate the possibility that such superstructures are a universal phenomenon of turbulence in general.

We hope that our argumentation above convinces the reviewer.

Yours sincerely, the authors.

Reviewer #2 (Remarks to the Author):

I have carefully read all three rebuttal files and the revised manuscript + supplement. The authors are to be commended for their systematic and thorough addressing of all comments. In particular, I find the response to Reviewer 3 comments to be persuasive. The authors indeed have achieved more than simply extending the parameter space of R-B simulations, through their careful analysis of spatial and temporal properties of the large-scale structures, and their relation to flow parameters and boundary-layer features. The work provides a broad view of trends in large-scale flow, and I recommend that it be published in Nature Communications in its present form.

Reviewer #3 (Remarks to the Author):

The authors made a good case in their rebuttal for why their paper should be published in Nature Communications and I do not want to oppose it. They also answered all my technical points, except for the one concerning the "erratic variations of temperature filaments". The argument based on "erratic variations of temperature filaments" is still not clear to me.

Second Response to Reviewer #3

Thank you once more for your comment.

The authors made a good case in their rebuttal for why their paper should be published in Nature Communications and I do not want to oppose it. They also answered all my technical points, except for the one concerning the “erratic variations of temperature filaments”. The argument based on “erratic variations of temperature filaments” is still not clear to me.

Answer: We have decided to take out this sentence. It was a qualitative description for the increasingly complex time dependence of the rolls as Ra grows starting from the weakly nonlinear regime. The slightly extended first paragraph of this section provides now a quantitative picture on the basis of our data.

We hope that we could address the point of the reviewer.

Yours sincerely, the authors.